# Non-Fullerene Small Molecule Electron-Transporting Materials for Efficient p-i-n Perovskite Solar Cells

**DOI:** 10.3390/nano10061082

**Published:** 2020-05-31

**Authors:** Da-Seul Choi, Sung-Nam Kwon, Seok-In Na

**Affiliations:** Graduate School of Flexible and Printable Electronics, LANL-JBNU Engineering Institute-Korea, Jeonbuk National University, 567 Baekje-daero, Deokjin-gu, Jeonju-si 54896, Korea; ektmf6175@jbnu.ac.kr

**Keywords:** perovskite solar cells, non-fullerene small molecules, electron transporting layer, lowest unoccupied molecular orbital energy level, charge extraction and recombination

## Abstract

PC_61_BM is commonly used in perovskite solar cells (PSC) as the electron transport material (ETM). However, PC_61_BM film has various disadvantages, such as its low coverage or the many pinholes that appear due to its aggregation behavior. These faults may lead to undesirable direct contact between the metal cathode and perovskite film, which could result in charge recombination at the perovskite/metal interface. In order to overcome this problem, three alternative non-fullerene electron materials were applied to inverted PSCs; they were evaluated on suitability as electron transport layers. The roles and effects of these non-fullerene ETMs on device performance were studied using photoluminescence (PL) measurements, field emission scanning electron microscopy (FE-SEM), atomic force microscopy (AFM), internal resistance in PSC measurements, and conductive atomic force microscopy (C-AFM). It was found that one of the tested materials, IT-4f, showed excellent electron extraction ability and was associated with reduced recombination. The PSC with IT-4f as the ETM produced better cell-performance; it had an average PCE of 11.21%, which makes it better than the ITIC and COi8DFIC-based devices. Finally, IT-4f was compared with PC_61_BM; it was found that the two materials have quite comparable efficiency and stability levels.

## 1. Introduction

Perovskite solar cells (PSCs) based on organo-metal halides have received a lot of attention recently due to their high efficiencies, facile solution processes, and low manufacturing costs [1,2,3]. The power conversion efficiency (PCE) of PSCs has increased rapidly and continuously since it was first reported in 2009; a PCE recently achieved 25.2% efficiency, which is comparable to silicon solar cells [4]. Hence, PSC is a highly promising candidate as an alternative energy source that can not only be commercialized through roll-to-roll processing, but can also prevent energy exhaustion and environmental pollution [2,5].

Currently, various PSC architectures are being actively studied; these include mesoporous structured, n-i-p structured, and p-i-n structured PCEs; over 20% efficiency has been gained using each architecture [4,6,7]. Although mesoporous-structure-based PSCs have shown quite high performances, the formation of mesoporous or compact TiO_2_ layers generally requires high thermal energy (>450 °C treatment), which limits continuous processes and scaled-up manufacturing for tandem and flexible PSC applications [8,9]. Regarding one of the other architectures, it has been reported that n-i-p structured PSCs, which use metal oxide as their electron transport layer (ETL), have hysteresis behavior [10,11,12]. Meanwhile, p-i-n structured PSCs, which are generally composed of a transparent conductive electrode/hole transport layer (HTL)/perovskite/ETL/metal electrode structure, are known to produce hysteresis-free PSCs through low-temperature and continuous processes [11,12,13,14].

In the p-i-n structured PSC, fullerene and its derivatives are widely used. In particular phenyl-C_61_-butyl ester (PC_61_BM) is commonly used as the ETL due to its high electron mobility and dissolution in non-polar solvents [13,14,15]. In addition, it has been demonstrated that PC_61_BM can provide a trap passivation function in the perovskite layer by interacting with negatively charged under-coordinated halide ions or Pb-I anti-site defects; this results in improved performance of the PSCs [16,17,18]. Despite these advantages, PC_61_BM ETL has some drawbacks to address; due to the low solubility and viscosity of the solution, and the cohesive behavior of PC_61_BM, it is difficult to form a uniform and defect-free PC_61_BM film on the perovskite layer [10,19]. Uneven and defective PC_61_BM films could cause unexpected direct contact between the metals and perovskite, resulting in recombination of photogenerated charges and degradation of device performance [8]. Moreover, mechanical analysis of the p-i-n PSC structure reveals that the PC_61_BM layer is the most fragile and susceptible to rupture. All the mentioned problems suggest that replacing PC_61_BM would lead to much enhanced device stability [14,20,21].

Fundamentally, to solve these problems completely, it is best to replace fullerene. Various studies have recently been conducted on the ETL in PSCs being composed of non-fullerene-based, small-molecule acceptor materials [14,22]. The motivation for these experiments could be because the fact non-fullerene small molecules (NFSMs) are easier and cheaper to synthesize than PC_61_BM, and the various properties of NFSMs can be also easily controlled by modification to the side chains [22]. For example, the small molecule 3,9-bis(2-methylene-(3-(1,1-dicyanomethylene)-indanone)-5,5,11,11-tetrakis(4-hexylphenyl)-dithieno[2,3-d:2′,3′-d′]-s-indaceno[1,2-b:5,6-b′]-dithiophene (ITIC), which is based on an indacenodithiophene (IDT) chemical backbone, has been used as an electron acceptor in organic solar cells, and these cells achieved comparable efficiencies with PC_61_BM [14,22,23]. ITIC has also been used to optimize the ETL of n-i-p structured PSCs; an ultra-thin ITIC layer was inserted between the TiO_2_ ETL and perovskite active layers to improve photovoltaic performance [24]. In our previous research, we also confirmed the ITIC’s feasibility as the ETL of PSCs based on using MAPbI_3_ as the photovoltaic active layer [22]. Meanwhile, Angmo et al. demonstrated that when industry-relevant slot-die coating methods are used, ITIC is a good candidate to replace PC_61_BM in ambient-processed PSCs. In fact, ITIC in nanofiber form improves device mechanical integrity when compared with the aggregated form of PC_61_BM film [14]. In this regard, various NFSMs based on IDT chemical backbone have been reported with a view of using them as PC_61_BM alternatives [25,26,27]. However, there have been few trials of these NFSMs being used in actual perovskite solar cells [14,16,22]. Thus, it is important that a practical comparison to confirm whether they are better ETL candidates be carried out.

In this respect, this work began as a study to solve the problems of PCBM and eventually find an electron transport material that can replace it. We first selected IDT-backbone-based materials as an alternative electron transport materials (ETMs); it was reported IDT-backbone-based materials have relatively good electron mobility and energy levels compared to other ETMs [25,26,27]. Three types of NFSMs with the same IDT backbone structure but different electronic mobility and energy levels were selected, and device performance changes were compared accordingly: ITIC, IT-4f (3,9-bis(2-methylene-((3-(1,1-dicyanomethylene)-6,7-difluoro)-indanone))-5,5,11,11-tetrakis(4-hexylphenyl)-dithieno[2,3-d:2′,3′-d’]-s-indaceno[1,2-b:5,6-b’]dithiophene), and COi8DFIC (2,2′-[[4,4,11,11-tetrakis(4-hexylphenyl)-4,11-dihydrothieno[2′,3′:4,5]thieno[2,3-d]thieno[2′′′′,3′′′′:4′′′,5′′′]thieno[2′′′,3′′′:4′′,5′′]pyrano[2′′,3′′:4′,5′]thieno[2′,3′:4,5]thieno[3,2-b]pyran-2,9-diyl]bis[methylidyne(5,6-difluoro)]]). These NFSMs were systematically compared for better performance using Cs_0.17_(FA_0.9_MA_0.1_)_0.83_Pb(I_0.87_Br_0.13_)_3_ (CsMAFA)-based perovskite solar cells. Considering the energy level and electronic mobility, IT-4F, which has a lower LUMO and better electronic mobility than ITIC and COi8DFIC, was expected to show the best device performance. Consequently, it was found that IT-4f showed excellent electron extraction ability and associated reduced recombination. PSCs with IT-4f as the ETM had a relatively high average PCE of 11.21% compared to ITIC and COi8DFIC-based devices, and showed comparable cell-performance with PC_61_BM-bsed devices.

## 2. Materials and Methods

For the fabrication of p-i-n structured PSCs with ETMs based on various NFSMs, firstly, pre-patterned indium tin oxide (ITO) glass-substrate (AMG-Tech) was treated with UV/O_3_ for 30 min. Next, for the HTL layer, NiO nanoparticle solution (NiO NPs, 2.5 wt% NiO ethanol, 1-material Inc., Dorval, QC, Canada) was spin-coated onto UV-treated ITO at 4000 rpm for 40 s and annealed at 350 °C for 30 min in air. In order to prepare the perovskite film, Cs_0.17_(FA_0.9_MA_0.1_)_0.83_Pb(I_0.87_Br_0.13_)_3_ (CsMAFA) perovskite precursor solution was prepared by dissolving 568 mg lead (ΙΙ) iodide (PbI_2_, 99%, TCI, Tokyo, Japan), 80 mg lead(II) bromide (PbBr_2_, 99%, TCI), 187 mg formamidinium iodide (FAI, 98%, Great Cell Solar, QLD, Australia), 66 mg cesium iodide (CsI, 99.99%, Sigma-Aldrich, St. Louis, MO, USA), and 12 mg methylammonium bromide (MABr, ≥99%, Sigma-Aldrich) in 0.8 mL of N,N-dimethylformamide (DMF, 99.8%, Sigma-Aldrich) and 0.2 mL of dimethyl sulfoxide (DMSO, 99.7%, Sigma-Aldrich) in an N_2_-filled glove box (<1 ppm O_2_ and H_2_O). The perovskite solution was spin-coated at 500 rpm for 5 s and 5000 rpm for 45 s in a two-step process. During the second coating step, 0.4 mL of chlorobenzene (CB, 99.8%, Sigma-Aldrich) was dropped on the spinning substrate 30 s before the end of the spin-coating; the film was then dried at 100 °C for 10 min. ITIC (1-material Inc.), IT-4f (1-material Inc.), and COi8DFIC (1-material Inc.) dispersed in 1,2-dichlorobenzene (DCB, 99%, Sigma-Aldrich) were spin-coated on the perovskite layer at 5000 rpm for 50 s, in which the thicknesses of NFSMs were optimized approximately 11.18 nm (5 mg/mL of ITIC), 10.10 nm (8 mg/mL of IT-4f), and 0.84 nm (8 mg/mL of COi8DFIC), respectively. PC_61_BM (Nano-C, Westwood, MA, USA) dispersed in DCB (20 mg/mL) was spin-coated on the perovskite layer at 5000 rpm for 50 s. A finger-shaped mask was accurately machined so that the vertically projected electrode and ITO overlap areas (active areas) were 0.0464 cm^2^. Then, using the corresponding mask, bathocuproine (BCP, 98%, Alfa Aesar, Ward Hill, MA, USA) (3 nm) and Ag electrodes (100 nm) were independently deposited to be separated from each other by a thermal evaporator under a pressure of 10^−6^ torr. Thus, as shown in Figure 1, the electrodes were configured independently to form one device, and statistical photovoltaic cell parameters were analyzed for each independent electrode.

The photocurrent density-voltage (J–V) characteristics of the PSCs were evaluated using a Keithley 2400 (Keithley Instruments Inc., Cleveland, OH, USA) and Oriel solar simulator (Sol3A, Class AAA, Newport, Irvine, CA, USA) under the AM 1.5 G (100 mW/cm^2^) with a standard Si-reference solar cell certified by the International System of Units (SI) (SRC 1000 TC KG5 N, VLSI Standards, Inc., Milpitas, CA, USA). Steady-state photoluminescence (PL), transient photovoltage (TPV), and transient photocurrent (TPC) measurements were performed using a spectrophotometer (SHIMADZU, Kyoto, Japan, RF-6000) and an organic semiconductor parameter test system (T4000, McScience, Suwon, Korea) with oscilloscope (DPO-2014B, Tektronix, Beaverton, OR, USA). The properties of film were examined by looking at surface images while checking its properties and work function using atomic force microscopy (AFM, XE7, Park System, Suwon, Korea), field emission scanning electron microscopy (FE-SEM, JSM-7100F, JEOL Ltd., Tokyo, Japan), ultraviolet photoelectron spectroscopy (UPS, Nexsa-XPS system, Thermo Scientific Ltd., Waltham MA, USA) with He I (21.2 eV) as the photon source, and conductive atomic force microscopy (C-AFM, Bruker, Billerica, MA, USA, MultiMode 8).

## 3. Results and Discussion

To investigate the impacts of the selected non-fullerene ETMs (ITIC, IT-4f, and COi8DFIC) on the performance of the perovskite solar cells (PSCs) and determine the optimum concentration of non-fullerene ETMs, we fabricated inverted planar PSCs with various concentrations of non-fullerene ETMs. Figure 1 illustrates the molecular structure of the selected non-fullerene ETMs and the device architecture of the inverted planar PSC: glass/ITO/HTL/perovskite/ETL/metal electrode. The non-fullerene ETM layers with various concentrations were spin-coated on the pre-prepared perovskite substrate, and the J–V curves of the PSC based on this are shown in Appendix A. The average device efficiency of the PSC with ITIC increased from 4.99% to 8.22% as the concentration of ITIC increased from 2 to 5 mg/mL, and decreased 5.16% as the concentration of ITIC further increased to 12 mg/mL. As shown in Appendix A, similar trends were observed for the J_SC_, V_OC_, and FF, for IT-4F and COi8DFIC. Moreover, similar trends were shown at slightly higher concentrations. Consequently, optimal device performance was obtained at ITIC of 5 mg/mL, IT-4f of 8 mg/mL, and COi8DFIC of 8 mg/mL, respectively.

To better study the impact of the selected non-fullerene ETMs on the performance of the PSC, we performed additional statistical analysis on more than 15 devices under optimal concentration conditions. Figure 2 shows the typical J–V curves and the statistical photovoltaic cell parameters of the PSCs with various ETMs under optimal concentration conditions. The detailed photovoltaic cell parameters are listed in Table 1. As can be seen in Figure 2a, the PSCs with ITIC and COi8DFIC showed relatively poor diode characteristics compared to the PSC with IT-4F. In addition, the PSCs with ITIC and COi8DFIC showed similar photovoltaic cell parameters to each other and similar average PCEs of 8.01% (ITIC) and 8.04% (COi8DFIC), as shown in Figure 2b,c, and Table 1. The PSCs with the IT-4f, in contrast, have better photovoltaic cell parameters, the PSCs with IT-4f exhibited an average V_OC_ of 0.94 V, an average J_SC_ of 19.12 mA/cm^2^, an average FF of 60.94%, and an average PCE of 11.19%. The best performing device with IT-4f exhibited a PCE of 12.11%, while ITIC and COi8DFIC exhibited quite comparable PCEs of 8.37% and 8.55%, respectively. It is clear to see that among the three non-fullerene ETMs, IT-4f had by far the best photovoltaic performance.

In order to gain some insight into the electron extraction properties of the various ETLs, steady-state photoluminescence (PL) quenching measurements were performed on the perovskite film with and without NFSMs-based ETLs. As shown in Figure 3a, the change in the PL intensity of the perovskite film according to the introduction of non-fullerene ETMs was measured. The PL intensities of the perovskite films with non-fullerene ETM coatings were significantly reduced compared to that of the pristine perovskite film. Additionally, it was clearly observed that the perovskite film with the IT-4f ETM had a low PL intensity compared to those of ITIC and COi8DFIC, which indicates that the IT-4f possesses relatively better electron-extraction ability from the perovskite [28]. To further study charge recombination and charge transfer dynamics in the inverted planar PSCs with non-fullerene ETMs, transient photovoltage (TPV) and transient photocurrent (TPC) measurements were performed [18,29]. The average TPV and TPC decay times were calculated by fitting TPV and TPC curves using a biexponential decay function [18]. Figure 3b and Appendix A show the average TPV decays of PSCs prepared with non-fullerene layers. It was found that the average TPV decay time and the charge recombination lifetime of the IT-4f-based device (649 µs) are longer than those of both ITIC and COi8DFIC-based devices, these devices having quite similar average TPV decay times (ITIC-based device (263 µs) and COi8DFIC-based device (257 µs)), as shown in Figure 3b. The prolonged charge recombination lifetime suggests that charge recombination is efficiently suppressed with the introduction of IT-4f. Meanwhile, the average TPC decay time of the IT-4f-based device (0.43 µs) was shorter than for ITIC and COi8DFIC, and the ITIC-based devices (0.55 µs) and COi8DFIC-based devices (0.56 µs) showed similar average TPC decay times, as shown in Figure 3c and Appendix A. This indicates that the device with IT-4f layers has more efficient charge extraction and charge transport capabilities [16,30,31]. These TPV and TPC results are comprehensively consistent with previous J–V and PL measurements. More importantly, these TPV and TPC results suggest that the improved photovoltaic performance of the IT-4f-based PSCs can be ascribed to more efficient charge extraction and reduced recombination at the IT-4f/perovskite interface compared to ITIC and COi8DFIC.

To find out the reason for the enhanced performance and charge extraction capabilities shown by the IT-4f-based device, we first studied the surface morphologies of the non-fullerene ETMs on the perovskite layer by scanning electron microscopy (SEM) and atomic force microscopy (AFM). Figure 4a shows the top-view SEM images of ITIC, IT-4f, and COi8DFIC films on a perovskite layer. These SEM images for the ITIC, IT-4f, and COi8DFIC on a perovskite layer all show a full coverage film without any distinguishable pinholes or defects. Figure 4b shows the AFM-based surface topographical images; it also shows the root mean square (RMS) roughness values of the ITIC, IT-4f, and COi8DFIC films coated on the perovskite layer, which are 9.83 nm, 11.47 nm, and 9.70 nm, respectively. After the introduction of NFSMs, the RMS roughness levels were all smoother compared to the pristine perovskite layer (16.99 nm). However, considering the ITIC and COi8DFIC showed similar RMS values and had better surfaces than IT-4f, and a smoother surface morphology could induce uniform contact between the perovskite layer and metal electrode for better charge transport and cell-efficiency [32,33]. This morphology analysis does not explain why the IT-4f-based device shows better performance. Next, since the energy level matching between the conduction band (CB) of perovskite and the lowest unoccupied molecular orbital (LUMO) of the ETL has a huge impact on PSC performance [28], based on previous reports, we investigated the energy level alignment between perovskite and NFSMs; the LUMO/HOMO values of the ITIC, IT-4f, and COi8DFIC are summarized as −3.89 (±0.042) eV/−5.48 (±0.014) eV, −4.14 (±0.035) eV/−5.66 (±0.078) eV, and −3.88 eV (±0.148)/−5.50 (±0.021), respectively [14,22,25,26,27]. Figure 4c presents the energy level diagram between the perovskite’s CB and the ETL’s LUMO and confirms that IT-4f provides better energy level matching with perovskite than ITIC and COi8DFIC, which could be favorable for charge extraction and cell-efficiency enhancement [28,34]. As shown in Figure 4d, similar trends were also confirmed by the series resistance (Rs) calculated from the J–V curve of the device manufactured under optimal concentration conditions; the same device used for the statistical data in Figure 2 was used. The average (Rs) value of IT-4f (2.73 ± 0.20 Ω cm^2^) was lower than those of ITIC (3.51 ± 0.10 Ω cm^2^) and COi8DFIC (3.39 ± 0.16 Ω cm^2^). Based on these observations, it is believed that the better cell-parameters obtained with IT-4f could be mainly due to the better energy-level alignment between the ETL and photo active layer that results in enhanced electron transport and PSC-efficiency. All these facts come together to confirm that IT-4f is a more promising candidate for use in high performance PSCs. This conclusion and the associated data are also supported by the conductive atomic force microscopy (C-AFM) that measures the current-flow between the sample and the AFM tip. As shown in Figure 4e, IT-4f has a higher average current (15.20 pA) than ITIC (4.32 pA) and COi8DFIC (3.02 pA); this indicates more efficient charge collection at the IT-4f ETL that might be caused by a lower injection barrier and a better ohmic contact [35,36,37,38,39].

To confirm the potential of IT-4F as an alternative electron transport material, we fabricated the device with PC_61_BM, commonly used as an electron transport material, and performed comparative analysis with the performance of IT-4F based devices. As can be seen from the statistical photovoltaic cell parameters for IT-4f and PC_61_BM-based devices shown in Figure 5a,b, IT-4f-based devices showed performance parameters similar to those of the PC_61_BM-based devices, resulting in an average PCE of 11.27%, similar to that of PCBM-based devices (11.57%). These results indicate that It-4F has potential as an alternative ETL. However, SEM and AFM analysis and energy level diagrams for PCBM suggest that there may be a trade-off between surface properties and energy levels of in the case of IT-4F. As shown in Appendix A, PCBM has a smoother surface (7.50 nm) than IT-4f (11.47 nm), which can lead to a relatively higher FF than for IT-4F-based devices [40,41]. On the other hand, as illustrated in Figure 4c and Appendix A, IT-4F has a relatively deeper and better matched LUMO level (−4.14 eV) than PCBM (−3.91 eV), and thus may exhibit more efficient charge collection ability, resulting in a relatively high Jsc compared to PCBM-based devices [40]. This can be supported by the results of the C-AFM of the IT-4F and PCBM, as shown in Figure 4e and Appendix A. Therefore, considering the surface roughness, it can be determined that the IT-4F-based device exhibits similar performance to the PCBM-based device due to the LUMO that is relatively well matched with CB of the perovskite. The stability of unencapsulated devices using IT-4f and PC_61_BM was also investigated because ETL in the p-i-n structure could have a significant impact on the stability of PSCs; corresponding results are shown in Figure 5c. The devices were stored indoors at ~23 °C under ~55% relative humidity in ambient air conditions prior to measurement. The efficiency of the IT-4f-based device was reduced to 55.9% after 664 h and that of the PC_61_BM-based device was reduced to 57.3% after identical exposure time; this demonstrates the quite similar stability of both devices. From this result, it can be implied that the IT-4f is a suitable material to be used as a PC_61_BM-alternative ETL in high performance PSCs. 

## 4. Conclusions

In summary, we systematically investigated the use of solution-processable NFSMs as ETLs for high-performance inverted PSCs. We introduced ITIC, IT-4F, and COi8DFIC for use as ETLs in inverted PSCs as part of a search for an improved alternative to PC_61_BM layers. It was found that IT-4f showed the best performance of all the investigated non-fullerene candidates. Various device and film-property analyses revealed that using IT-4f, the device showed superior charge extraction and enhanced suppressed charge recombination, and the energy-level alignment was better matched with the perovskite, resulting in improved photovoltaic performances. Furthermore, when comparing the IT-4f device with the widely used PC_61_BM device, it was found that the efficiency and stability of the IT-4f based PSC (11.75%) are quite alike those of the PC_61_BM-based PSCs. Our future work will focus on finding other non-fullerene ETLs with the aim of comprehensively improving on the present PC_61_BM-alternatives to produce even higher performance PSCs.

## Figures and Tables

**Figure 1 nanomaterials-10-01082-f001:**
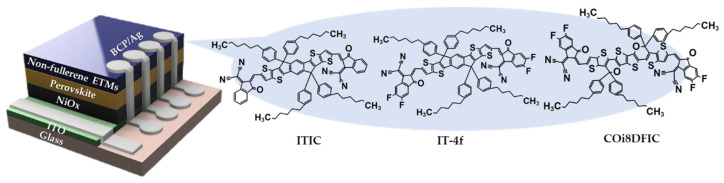
Schematic illustration of the molecular structure of the selected electron transport materials (ETMs) and the device architecture of the inverted, planar, non-fullerene perovskite solar cells (PSC).

**Figure 2 nanomaterials-10-01082-f002:**
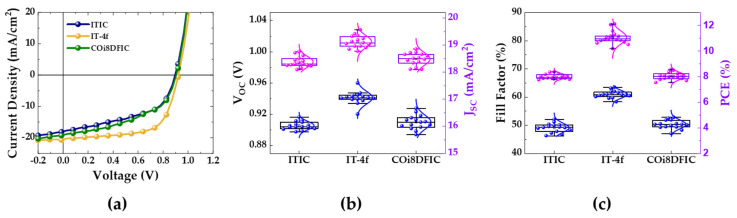
(**a**) Typical current density–voltage (J–V) curves and (**b**,**c**) statistical photovoltaic cell parameters of the PSC with various ETMs under optimal concentration conditions.

**Figure 3 nanomaterials-10-01082-f003:**
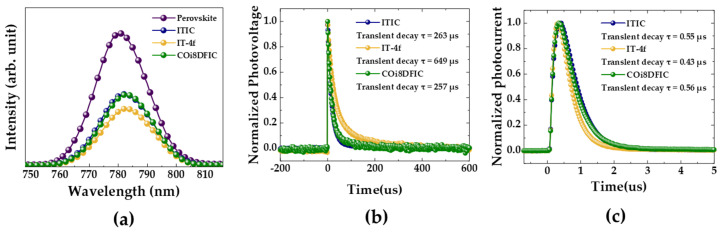
(**a**) Steady-state photoluminescence (PL) spectra of the perovskite, perovskite/ITIC, perovskite/IT-4f, and perovskite/COi8DFIC. (**b**) Transient photovoltage (TPV) and (**c**) transient photocurrent (TPC) curves of the PSC with ITIC, IT-4f, and COi8DFIC as ETLs.

**Figure 4 nanomaterials-10-01082-f004:**
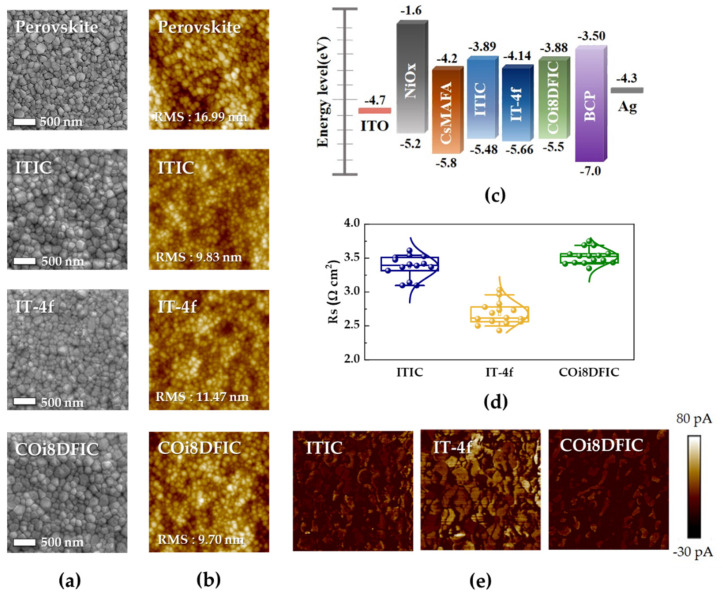
(**a**) Top-view SEM and (**b**) AFM images (scale: 5 × 5 µm) of ITIC, IT-4f, and COi8DFIC films coated on the ITO/NiO_x_/perovskite layer. (**c**) Schematic energy level diagram of the perovskite and non-fullerene ETMs. (**d**) Series resistance (Rs) calculated from the J–V curve of the device manufactured under optimal concentration conditions. (**e**) C-AFM images (scale: 2 × 2 µm) of ITIC, IT-4f, and COi8DFIC films.

**Figure 5 nanomaterials-10-01082-f005:**
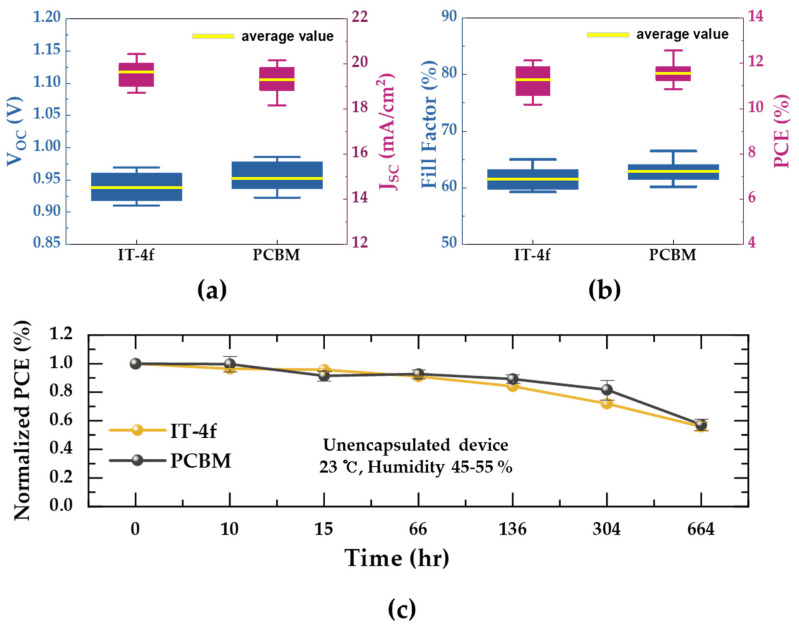
(**a**,**b**) Statistical photovoltaic cell parameters of the PSC with IT-4f and PCBM. (**c**) Changes in power conversion efficiency of the PSC with IT-4f and PCBM.

**Table 1 nanomaterials-10-01082-t001:** Summary of statistical photovoltaic cell parameters of the PSC with various ETMs under optimal concentration conditions.

Sample	V_OC_ (V)	J_SC_ (mA/cm^2^)	FF (%)	PCE (%)	PCE_MAX_ (%)
ITIC	0.91 (± 0.01)	18.38 (± 0.20)	49.06 (± 1.61)	8.01 (± 0.19)	8.37
IT-4f	0.94 (± 0.01)	19.12 (± 0.22)	60.94 (± 1.33)	11.19 (± 0.40)	12.11
COi8DFIC	0.91 (± 0.01)	18.48 (± 0.23)	50.25 (± 1.54)	8.04 (± 0.29)	8.55

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
