# Peer review of "Non-Fullerene Small Molecule Electron-Transporting Materials for Efficient p-i-n Perovskite Solar Cells"

_nanomaterials, 2020, doi:10.3390/nano10061082_

Round 1
Reviewer 1 Report
The paper reports a detailed investigation on non-fullerene materials used to enhance PSC characteristics. The paper is well written, in a clear form, but some corrections has to be done, especially on punctuation (attached pdf highlights some parts) and few words.
In section 2, authors should specify top Ag electrode geometry. They only talk about a 4.64 mm2 mask, whereas it is not indicated if they used a single contact (fingers of fig. 1 conncted in parallel) or if different fingers are used for a “statistical” analysis on the same sample. In this way they must explain how “statistical photovoltaic cell parameters” have been acquired. Moreover, data of figs. 2d and 2e refer to the same concentration for each material?
It is not clear the meaning for the errors reported in tab. 1 (for example, what’s the meaning of “0.906 (max 0.934) (+/- 0.02)”): results reported in fig. 2e and 2f show much higher “dispersion”.
Results depicted in fig. 3a do not display a so high quenching ability of IT-4f as authors highlight. In my opinion, PL results have to be moderately underlined.
Transient photovoltage signal for IT-4f shown in fig. 3b displays a much longer decay time (of the order of 50-100 us) than the authors stated (21.9 us). Is it a typos error or authors just considered only one of the two exponential contribution? Why don’t authors use a “decay time” given by the time required for the signal to go from 90% to 10% of its dynamics? Moreover, as done in ref. 19, they should use the “average decay time” as defined in the reference.
Please, correct labels in fig. 4.
About roughness. Images reported in fig.4a display “grains dimension” of about 5-10 um. How can roughness as low as tens of nm? In addition, AFM images clearly show the presence of “deep holes” on the surface. How did authors calculated the rms value for the roughness?
Also, for data reported in fig. 4d it is not clear how the statistics was performed. Again, authors should specify how many samples they characterized (same concentration based on results of fig. 1a-c?)
However, due the huge dispersion, Rs appears of the same order for all the investigated materials and authors are not encuraged to stress too much on such a result. More impressive are results obtained with C-AFM technique.
Finally, the number of references appears excessive compared to the content of the article. Authors are encuraged to limit their number.

Author Response
We appreciate the referee’s time and efforts in improving the manuscript. We revised the manuscript according to the referee’s comments and used Microsoft Word's "Track changes" feature to highlight the changes in the manuscript.
Please see the attachment.

Reviewer 2 Report
see attached report

Author Response

(The authors gave the same response as above.)

Reviewer 3 Report
Please find attached.

Author Response

(The authors gave the same response as above.)

Round 2
Reviewer 1 Report
Dear authors,
thank you for providing modification suggested in the previous submission.
Please find attached your submitted pdf where some suggestions related to a couple of sentences to be reworded, in my opinion, as well as some indications of typos errors to be corrected, are provided as comments.
Regards

Author Response
We appreciate the careful review and constructive suggestions.
Based on the reviewer's comments, we have modified the manuscript again so that each part can convey its meaning more clearly.
We also have used Microsoft Word's "Track changes" feature to highlight the changes in the manuscript.
It is our belief that the manuscript has greatly improved after the proposed editing.

Reviewer 2 Report
The Authors have addressed all the issues raised in the first round of the peer review.
I can now recommend publication of the manuscript in the present form.
Author Response
We thank you for your careful review and constructive suggestions.
Your dedication has made our manuscript even better.
Reviewer 3 Report
All issues have been resolved.
Author Response

(The authors gave the same response as above.)
